# Mining and Metallurgical Waste as Potential Secondary Sources of Metals—A Case Study for the West Balkan Region

Robert Šajn [1,*], Ivica Ristović [2] and Barbara Čeplak [1]

1    Geological Survey of Slovenia, 1000 Ljubljana, Slovenia; barbara.ceplak@geo-zs.si
2    Faculty of Mining and Geology, University of Belgrade, 11030 Beograd, Serbia; ivica.ristovic@rgf.bg.ac.rs
*    Correspondence: robert.sajn@geo-zs.si

**Abstract:** The aim of this paper is to present a chemical composition and quantities of mining and processing waste landfills material developed during historical mining and smelting. After detailed inspection, it was found that approximately 2.6 gigatons of the waste had been deposited at 1650 sites, covering almost 65 km$^2$. More than half of this material, 55%, is characterized as conventional mining waste, 37% belongs to the processing tailings, and 8% to metallurgical waste. Most of these tailing sites are unclaimed, presenting a source of contamination for nearby communities. According to the literature data collected and additional chemical analyses, in accordance with zero-waste philosophy, about 42 promising locations (c. 270 million tons) could be selected, where various advanced eco-innovative methods of recovery could possibly apply. The areas with the highest prospective recovery are Serbia and Kosovo. In accordance with the metal prices achieved in March 2022, it is estimated that the recovery of tailings could bring up to 18,100 million USD, which is much more compared to the prices of March 2020—10,600 million USD—when the commodity market was governed by the COVID-19 restrictions. In addition to the commercial value of the metals, the environmental aspect should not be forgotten after the application of reuse and recycling concepts.

**Keywords:** secondary deposits; West Balkan; valuable elements; economic prospective metal recovery



## 1. Introduction

Possible political instabilities in countries and unforeseen situations such as the COVID-19 pandemic can lead to a rapid and deep economic crisis and seriously threaten the import of goods, including strategically important raw materials, from abroad to EU countries. Given the current political developments at the global level, self-sufficiency in strategic metals and other mineral resources has become an even more important issue for decision-makers in European countries. Indeed, the industry of the countries of the European Union is heavily dependent on imports of raw materials from abroad [1]. With the trend toward decarbonization and the global energy transition, the demand for raw materials to produce wind turbines, batteries, robots, etc., has increased dramatically. For example, a total of 44 raw materials are associated with the production of robots, and the EU is fully dependent on the supply of 33 of these raw materials. Another example: the production of fuel cells requires about 30 raw materials, 13 of which are on the EU list of critical elements [2]. South America, especially Chile and Peru, are the main source countries for the world's copper, while Brazil and Chile are the main source regions for the EU's iron and lithium, respectively. South Africa is a major supplier of platinum elements, while West Africa and Indonesia are the main source regions for bauxite and tin, respectively [3].

The European Union dependence on the mineral raw materials supply is showing a vulnerability to supply chain disruption. Imports of goods from abroad to the European Union countries have doubled since 2002, according to EUROSTAT trade data [4]. In 2021, China accounted for 22% of total imports [4]. In addition, primary mineral deposits are mostly depleted, and the remaining ore bodies have lower metal content, which may

lead to economically unviable exploitation, or exploitation requiring new state-of-the-art technologies that would enable economic recovery of metals [5]. Another key point: long-term mining and metallurgical activities have left a huge amount of waste material. Globally, it is estimated that the annual production of solid waste from primary production of mineral and metallic raw materials is over 100 billion tons [6]. Although this waste poses a threat to the environment [7], it can still contain a significant number of valuable elements [8]. Moreover, reprocessing of tailings can transform a linear economy into a circular economy, which is important to achieve the Sustainable Development Goals [9], while reducing dependence on reserve extraction [10]. There are several studies worldwide that examine reprocessing waste materials to find new sources of metals. For example, the United States has discovered saline water as a source of critical metals and is introducing so-called aqueous mining as an alternative to traditional hard rock mining [11]. South Africa is rich in platinum group metals (PGMs); however, low concentrations of these valuable metals are still found in the waste generated during ore mining. The study concluded that the process of phytomining could be a promising tool for the recovery of these valuable metals [12]. The importance of mineral supply was also highlighted in the study by Zglinicki et al. [13]. In fact, the survey conducted during the pandemic of COVID-19 shows that tailings piles in Indonesia can be a potential new source of critical raw materials. In addition to the economic aspects and supply of raw materials, several studies have addressed the environmental impacts of long-term mining and metallurgical activities. For instance, the increasing exploitation of natural resources in Saudi Arabia has undoubtedly left behind large amounts of waste that can damage the environment. The study by Hefni et al. [14] presents mining processes that enable smooth operating and in this way not to hold back the country's economic growth while being sustainable. Similarly, the study by Rybak et al. [15] based on mining in the Ural region describes the process of preserving valuable metals and using the remaining waste as backfill to prevent the possible collapse of tailings piles.

Recognized by the European Commission one of the possible regions to reduce Europe's dependence on imported mineral resources is the West Balkan [16]. West Balkan countries include Albania, Bosnia and Herzegovina, Kosovo (* This designation is without prejudice to positions on status, and is in line with UNSC 1244 and the ICJ Opinion on the Kosovo declaration of independence), North Macedonia, Montenegro, and Serbia [17]. The countries of the West Balkan have stagnated or even regressed economically due to political developments in the previous century, but owing to their long history of mining, processing, and production of raw materials, they have enormous potential for metal recovery from so-called SRMs—secondary raw material deposits, which could additionally improve their socioeconomic position. In the 20th century, mining played a key role in the former Yugoslavia and Albania, but the fiasco of the Yugoslavia market led to a deterioration of economic conditions in the region with a sharp decline in the early 1990s [18]. As a result, the closure of operations such as mining was initiated, resulting in thousands of old abandoned mining sites. However, the rapid and improper closure and the lack of a legal owner to prevent or minimize the risk of accidents and pollution further complicated the rehabilitation of these tailing piles [18].

Based on previous studies, mine wastes in the West Balkan were found to be rich in gold, silver [19], zinc, lead [20], and copper [21]. Gold, silver, and copper are so-called precious metals because they occur naturally in native form. Gold and silver were used as monetary metals until the 1970s. They are commonly used in jewelry, dentistry, medicine, and the electrical industry. Gold and its alloys are used in corrosion-resistant alloys for chemical process equipment and in rupture disks [22], while silver is additionally used for photographic material, bearings, brazing and soft solders, catalysts, batteries, mirrors, etc. [23]. Copper is widely used in architecture due to its excellent corrosion resistance and long service life. Copper is also the best electrical conductor after silver and is widely used for energy efficient circuits and in radiators, automobiles, watches, computers, and telecommunications, etc. [24]. The other important metals for the world economy are lead

and zinc. Lead is mainly used in production of lead–acid batteries, cast metals (engines and appliances), and lead sheets (construction, medical radiation shielding) [25], while zinc is mainly used for galvanizing, brass and bronze, zinc-based paints, etc. [26]. The element that also plays an important role in manufacturing is molybdenum. Recently, its utility has gained importance in green technology, especially in the production of biofuels, catalysts, ethanol, solar cells, and wind turbines [27].

The purpose of this paper is to present the chemical composition of mining waste material in the West Balkan as possible sources of strategically important metals. This contribution uses and disseminates the data presented in the not yet published article by Šajn et al. [28] and for the first time shows the total profitability of metal recovery in the West Balkan. It presents the most promising sites for the recovery of metals from mine tailings in the West Balkan and estimates, based on the contents and quantities of certain elements in the tailings and the size of the deposits, the economic value of their possible future recovery. The study does not give a total cost needed for metal recovery but only estimates the potential yield. The article presents the study of the metal prices in 2000, 2020, and 2022, and lastly shows the final value that would be obtained by extracting metals from the West Balkan tailings. In addition to the economic aspect, the paper also highlights the environmental and socioeconomic aspects of the region, which could result from successful rehabilitation of the tailings.

## 2. Materials and Methods

### 2.1. Study Area

The West Balkan (WB) region consists of six countries: Serbia, Albania, Bosnia and Herzegovina, Croatia, Kosovo *, Montenegro, and North Macedonia. The terrain of the region is complex and spreads over hilly and mountain regions on the south and west, foreshore area on the east, and over a low altitude of the Pannonia Valley on the north, covering an area of 250,000 km$^2$ (Figure 1) [29]. The weather in the peninsula is changeable and is greatly influenced on the one hand by the sea, and on the other by the mountains [29]. The northern and central parts of the Balkans are characterized by a central European climate, resulting in cold winters, warm summers, and well-distributed precipitation. The southern and coastal areas, however, have a Mediterranean climate with dry and hot summers and mild and rainy winters [30].

The area of the West Balkan is struggling with severe depopulation. The area has about 16 million inhabitants, and this number is decreasing every year. Bosnia and Herzegovina have 27% fewer inhabitants today than in 1990, while the figures for Croatia, Albania, Kosovo *, and Serbia are 15%, 14%, 5%, and 9%, respectively. In contrast, Montenegro and North Macedonia have about 4% more inhabitants today than in 1990 [31]. In general, the fertility rate is low, less than 1.5 children per couple, which means a 10% decrease in population [32]. In addition to declining birth rates, the region suffers from conflict-induced migration and economic hardship [32]. It is estimated that by 2050, the countries will have between 20% and 40% fewer people than in 1990 [33]. The gross domestic product (GDP) for 2020 in this region ranged from approximately 5 billion USD for Montenegro to 57 billion USD for Croatia, while in 2020 the gross domestic product purchasing power parity (GDP PPP) ranged from approximately 20 billion USD for Kosovo * to 117 billion USD for Croatia [31].

Geologically, the study area belongs to the Alpine–Balkan–Carpathian–Dinaride belt, which formed in the Middle to Late Cretaceous and Late Eocene–Oligocene as a result of two independent phases of continent–continent collision [34]. The interesting geological structure of the region led to the formation of several mineral deposits. The territory of Albania is rich mainly with chromium, together with copper, nickel, and iron. Chromium is mined and processed in the Dibra and Mat districts, copper in Mirdite and Puke, and iron–nickel in Pogradec. The largest copper mining and processing complex is Fushe Arrez, while copper concentrations were smelted in Kukes, Lac, and Rubic [18]. The main plant for the production of ferronickel and ferrochromium–nickel and ferrochromium is located

in Elbasan. In addition to these metals, petroleum and lignite are also among Albania's important raw materials [35].

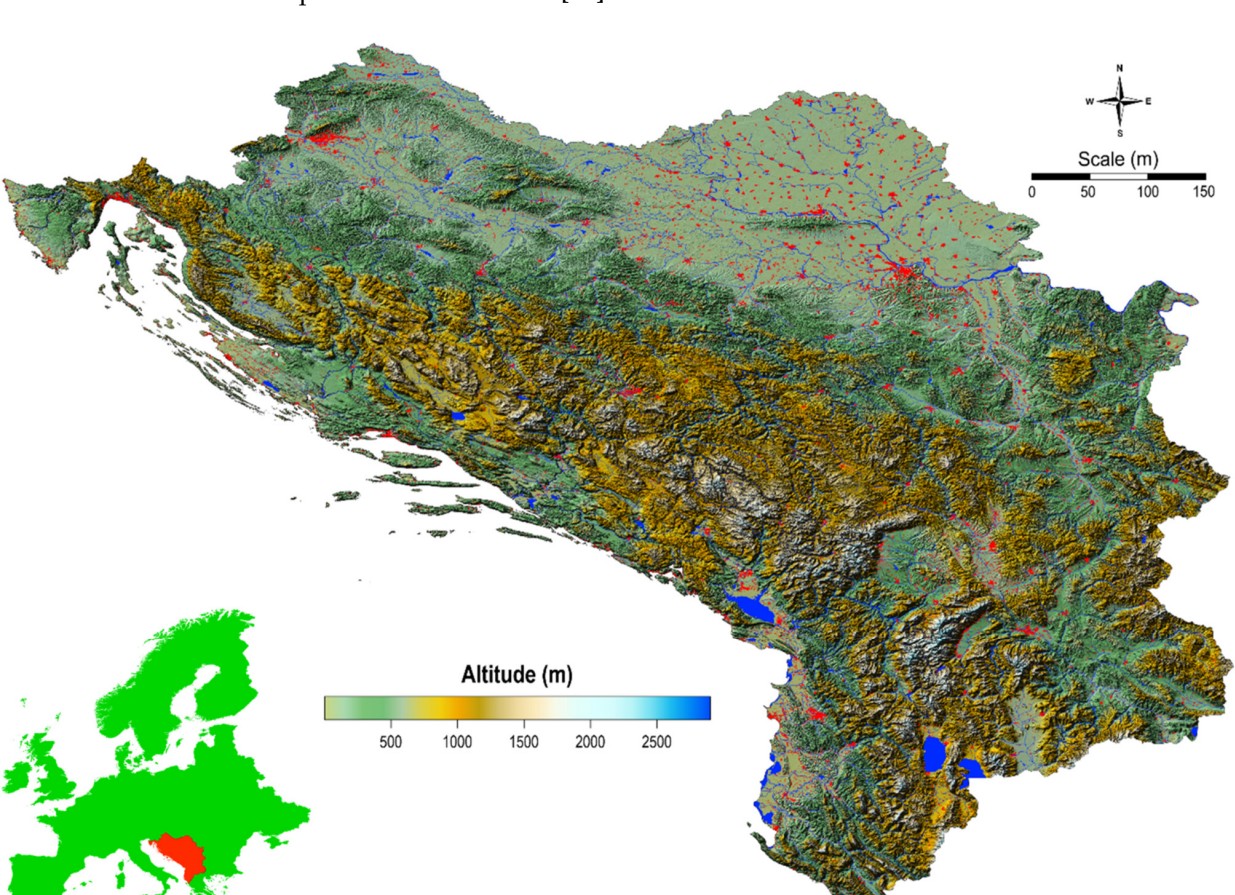

**Figure 1.** Digital elevation model of the West Balkan region.

Before the dissolution of the Federal Republic of Yugoslavia, Bosnia and Herzegovina were an important center for metallurgical industries, inter alia, steel output Rudarsko Metalurški Kobinat plant at Zenica and iron smelting in Vareš, while Srebrenica is important for lead and zinc mining. The country is also rich in bauxite, alumina, and aluminum [18]. Montenegro is important due to its lead and zinc tailings storage facility Mojkovac and Gradac, while economic concentrations of lead and zinc were proved in the mountain regions Ljubisnja (Šuplja Stijena) and Bjelasica (Brskovo Mine). Additionally, the Podgorica aluminum plant [18] and the ironworks in Nikšić are of great importance to the country. Montenegro is also rich in coal; thermal landfill slag and ash are stored in Maljevac [36].

North Macedonia has economic grades of copper, iron, lead, and zinc, together with precious metals such as silver and gold. The industrial sector contributes approximately 32% of GDP, while the agricultural sector 19% [37]. The most important mines in the country follow: the copper mine in Bučim; lead and zinc mines in Sasa, Zletovo, and Toranica; and the chromium and antimony mine in Lojane. There is also the smelter and refinery to produce lead and zinc and associated metals in Veles, ironworks in Skopje, and ferroalloy plants at Tetovo and Kavadarci [37]. As is the case for the other West Balkan countries, mining and mining-related activities represent an important part of Serbia's economy too. Namely, Serbia is rich in copper, coal, lead, and zinc, with associated gold, silver, copper, bismuth, and cadmium admixture. The most important mines in Serbia are copper ore deposits in Bor, Majdanpek, and Veliki Krivelj. In addition to base metals, precious metals are also detected in the Bor metallogenic zone (e.g., gold and silver). A high concentration of gold was also found in Lece. Other important sites in Serbia are a lead

smelter in Zajača, the coal mining and smelting complex at Kolubara, and ironworks in Smederevo [18].

The most important mineral resource in Kosovo * is lignite. The country has the fifth largest proven lignite reserves in the world [38]. In 2018, more than 7 million tons of lignite were mined [35]. The most important deposits are in Dukagjini and Drenica. Of particular importance in Kosovo * is the Trepča complex, where lead–zinc and silver have been mined since pre-Roman times. In addition, Kosovo * is rich in nickel and cobalt (e.g., mining district in Glavica), while the most important plants to produce ferronickel and ferrochrome are located in Glogovac. There are also deposits of iron (e.g., in the Glogovac mining district), chromium (e.g., in the Djakovica mine), and bauxite (e.g., in the Grebnik mine) [38].

*2.2. Data Collection*

The countries of the West Balkan have the potential to extract raw materials from primary and secondary raw material deposits and can increase their self-sufficiency in raw materials by using appropriate, modern, and eco-efficient technology. The data presented in this study were partly obtained during the period of European projects funded by EIT Raw Materials, such as RESEERVE [39], RIS-CuRE [40], and RIS-RECOVER [41], where some authors also participated, partly from the authors' archive documents, which are presented to the public for the first time as well as from the report [19] and the article [42].

During the duration of RESEERVE project, data concerning 1460 deposits at 113 sites were obtained. Data were collected from flotation tailings, red mud dams, and metallurgical slags. The further segmentation is shown in Figure 2. However, it was concluded that not all deposits represent important sources for the possible recovery of raw materials in the future. Therefore, sites that did not meet the previously established criteria (tailings size must be greater than 0.5 million tons, with a promising chemical composition in an inactive, abandoned, or nonremediated state, tailings must be the result of older flotation methods and allow for the possibility of recovery methods) were excluded from the survey. In the end, 34 waste sites remained; however, this study dealt only with 17 of the best prospective locations obtained during the RESEERVE project. The process of data collection to form the mineral register of the West Balkan within the RESEERVE project, disseminated in this study, is described in detail in the not-yet-published article by Šajn et al. [28].

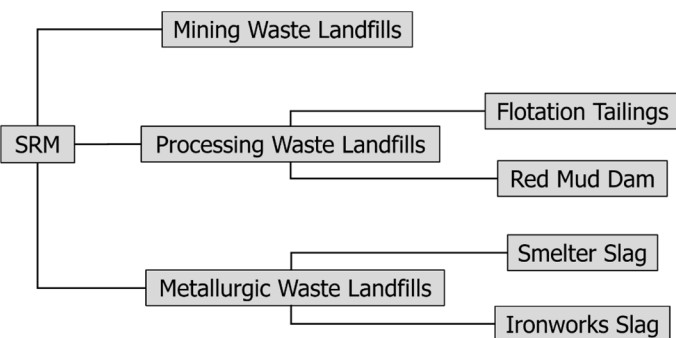

**Figure 2.** Types of mining, processing, and metallurgic waste.

In addition to data from the RESEERVE project, this article also uses data from the RIS-CuRE and RIS-RECOVER projects. Thus, to the 17 sites potentially defined during the RESEERVE project, 12 sites were added from which data were obtained during the life of the RECOVER and RIS-CuRE projects [40,41]—more precisely, data from hand and machine drilling. To discuss the prospects for metal recovery in the wider area of the West Balkan, the following data were added: (i) one site according to the data of the report [42], (ii) 9 sites according to the archival material from Kosovo * and by the authors, and (iii) three from the report by BRGM [19]. In the end, it was concluded that the area of the West Balkan consists of 42 potential waste disposal sites. The surface area and quantity of stored material were

first estimated based on data from the Google Earth app. The data were then digitized while further calculations were performed using the QGIS program.

*2.3. Laboratory Analysis*

This article combines information on tailings from several sources; thus, there is not a uniform sampling technique. To obtain a representative sample, surface samples (in Tables 1–3, remarked with 1, 4, and 6) were collected from at least five different microlocations on the tailing noted. Subsequently, they were mixed and stored in a plastic bag. Samples obtained during hand and machine drilling were taken at different depths, creating a representative sample (in Tables 1–3, remarked with 2 and 3). The remaining information concerning the samples was obtained based on the results presented in the report by Grieco et al. [42] and by BRGM [19], or by the authors' personal archive (in Tables 1–3, remarked with 5, 7 and 8).

**Table 1.** Basic data of potentially prospective SRM deposits.

| Location | Remark | Country | Deposit Type | Deposit Status | Main Elements | Surface (ha) | Quantity (Mt) | Element |
|---|---|---|---|---|---|---|---|---|
| Fushë Arrëz | 8 | Albania | FT | Abandoned | Cu | 19 | 3.1 | Au, Cu |
| Kukës | 4 | Albania | SSL | Abandoned | Cu | 13 | 3 | Cu |
| Rehove | 4 | Albania | FT | Abandoned | Cu | 4 | 0.6 | Cu |
| Reps | 4 | Albania | FT | Abandoned | Cu | 5 | 3.7 | Cu |
| Rreshen | 4 | Albania | FT | Abandoned | Cu | 3 | 0.5 | Cu |
| Sase I | 1 | B and H | FT | Abandoned | Pb/Zn | 6 | 1 | - |
| Sase II | 1 | B and H | FT | Active | Pb/Zn | 4 | 1 | Zn |
| Veovača | 1 | B and H | FT | Abandoned | Pb/Zn | 6 | 2 | Ag |
| Badovac | 6 | Kosovo * | FT | Abandoned | Pb/Zn | 9 | 8 | Ag, Au, Pb |
| Kišnica | 5 | Kosovo * | FT | Abandoned | Pb/Zn | 49 | 2 | Ag, Au, Bi, Pb, Zn |
| Leposavić I | 6 | Kosovo * | FT | Abandoned | Pb/Zn | 12 | 2.7 | - |
| Leposavić II | 6 | Kosovo * | FT | Active | Pb/Zn | 21 | 3.7 | - |
| Novo Brdo | 5 | Kosovo * | FT | Abandoned | Pb/Zn | 3 | 0.5 | Ag, Au, Pb, Zn |
| Gornje polje I | 6 | Kosovo * | FT | Abandoned | Pb/Zn | 26 | 9 | Au |
| Gornje polje II | 6 | Kosovo * | FT | Abandoned | Pb/Zn | 32 | 11 | Ag, Au, Bi, Pb, W |
| Gornje Polje III | 6 | Kosovo * | SSL | Abandoned | Pb/Zn | 6 | 2.5 | Ag, Bi, Cu, Pb, Sb, Zn |
| Žarkov potok | 5 | Kosovo * | FT | Abandoned | Pb/Zn | 23 | 10 | Ag, Au, Bi, Zn |
| Žitkovac | 6 | Kosovo * | FT | Abandoned | Pb/Zn | 28 | 12 | Ag, Au, Bi, Pb, W |
| K. Mitrovica | 6 | Kosovo * | SSL | Abandoned | Pb/Zn | 29 | 3 | Ag, Au, Bi |
| Gradac | 3 | Montenegro | FT | Abandoned | Pb/Zn | 13 | 3 | - |
| Probištip I | 2 | N. Macedonia | FT | Abandoned | Pb/Zn | 24 | 7 | Ag, In, Pb, Zn |
| Probištip II | 2 | N. Macedonia | FT | Abandoned | Pb/Zn | 40 | 12 | Ag, Au |
| Probištip III | 2 | N. Macedonia | FT | Active | Pb/Zn | 33 | 11 | - |
| Sasa I | 2 | N. Macedonia | FT | Abandoned | Pb/Zn | 12 | 5 | - |
| Sasa II | 2 | N. Macedonia | FT | Abandoned | Pb/Zn | 12 | 5 | - |
| Sasa III | 2 | N. Macedonia | FT | Active | Pb/Zn | 23 | 10 | Pb |
| Toranica | 2 | N. Macedonia | FT | Active | Pb/Zn | 14 | 10 | Pb |
| Veles | 3 | N. Macedonia | SSL | Abandoned | Pb/Zn | 4 | 1.8 | Ag, Cu, In, Pb, Zn |
| Lojane | 2 | N. Macedonia | FT | Abandoned | Sb | 2 | 0.5 | Sb |
| Bor I | 3 | Serbia | FT | Abandoned | Cu | 18 | 8 | Au, Cu |
| Bor II | 3 | Serbia | FT | Abandoned | Cu | 37 | 22 | Au, Cu |
| Bor III | 1 | Serbia | FT | Active | Cu | 71 | 50 | Cu, Mo, Re, Zn |
| Bor IV | - | Serbia | SSL | Active | Cu | 19 | 15 | - |
| Bor V | - | Serbia | SSL | Active | Cu | 9 | 2 | - |
| Grot | 1 | Serbia | FT | Active | Pb/Zn | 26 | 5.5 | - |
| Lece | 7 | Serbia | FT | Active | Pb/Zn | 23 | 2.7 | Au, In, Zn |
| Rudnica I | 1 | Serbia | FT | Abandoned | Pb/Zn | 11 | 3 | Cu |
| Rudnica II | - | Serbia | FT | Abandoned | Pb/Zn | 9 | 2.5 | - |
| Rudnik | 1 | Serbia | FT | Active | Pb/Zn | 41 | 8.7 | Ag, Pb |
| Veliki Majdan I | - | Serbia | FT | Abandoned | Pb/Zn | 2 | 0.8 | - |
| Veliki Majdan II | - | Serbia | FT | Active | Pb/Zn | 3 | 1.1 | - |
| Zajača | 1 | Serbia | SSL | Closed | Sb | 5 | 0.6 | Ag, Sb |

(*)—"This designation is without prejudice to positions on status and is in line with UNSC 1244 and the ICJ Opinion on the Kosovo declaration of independence.", B and H—Bosnia and Herzegovina, FT—flotation tailings, SSL—smelter slag landfill, Mt-million tons. Data source: 1—surface sampling (RESEERVE), 2—hand drilling/trenching, 3—machine drilling, 4—data from Albanian geological survey, 5—BRGM report [19], 6—surface sampling/archive material—Ristović, I., 7—personal archive (I. Ristović), 8—data from the article by Grieco et al. [42].

**Table 2.** Content of important chemical elements according to SRM landfills.

| Location | Remark | Ag (g/t) | Au (g/t) | Bi (g/t) | Cu (%) | In (g/t) | Mo (g/t) | Pb (%) | Re (g/t) | Sb (g/t) | W (g/t) | Zn (%) |
|---|---|---|---|---|---|---|---|---|---|---|---|---|
| Fushë Arrëz | 8 | 1.6 | 0.79 | - | 0.18 | - | 9.0 | <0.10 | - | 6.1 | - | <0.10 |
| Kukës | 4 | - | - | - | 0.17 | - | - | - | - | - | - | - |
| Rehove | 4 | - | - | - | 0.15 | - | - | - | - | - | - | - |
| Reps | 4 | - | - | - | 0.18 | - | - | - | - | - | - | - |
| Rreshen | 4 | - | - | - | 0.18 | - | - | - | - | - | - | - |
| Sase I | 1 | 6.3 | 0.09 | 3.6 | <0.01 | 1.3 | 1.2 | 0.27 | <0.01 | 150 | 16 | 0.27 |
| Sase II | 1 | 4.1 | 0.04 | 1.3 | <0.01 | 1.3 | 1.1 | 0.23 | <0.01 | 260 | 6.9 | 0.51 |
| Veovača | 1 | 33 | 0.07 | <1.0 | 0.02 | <0.10 | 3.2 | 0.36 | <0.01 | 240 | 1.0 | 0.46 |
| Badovac | 6 | 15 | 0.57 | 22 | 0.02 | 6.6 | <1.0 | 0.59 | <0.01 | 66 | 31 | 0.32 |
| Kišnica | 5 | 26 | 2.3 | 52 | 0.06 | 6.6 | 8.0 | 0.59 | - | 120 | 16 | 1.4 |
| Leposavić I | 6 | 6.8 | 0.09 | 7.4 | 0.01 | 1.8 | 1.4 | 0.39 | <0.01 | 110 | 8.5 | 0.16 |
| Leposavić II | 6 | - | - | - | - | - | - | - | - | - | - | - |
| Novo Brdo | 5 | 19 | 2.5 | 44 | 0.04 | 3.9 | 8.0 | 0.52 | - | 120 | 10 | 1.1 |
| Gornje polje I | 6 | 5.0 | 0.95 | 24 | 0.05 | 2.4 | <1.0 | 0.21 | <0.01 | 170 | 88 | 0.17 |
| Gornje polje II | 6 | 13 | 0.82 | 64 | 0.02 | 2.6 | 1.4 | 0.60 | <0.01 | 330 | 120 | 0.29 |
| Gornje Polje III | 6 | 86 | 0.06 | 77 | 0.27 | 6.8 | 38 | 5.1 | <0.01 | 1100 | 47 | 8.0 |
| Žarkov potok | 5 | 10 | 0.21 | 60 | 0.05 | 3.3 | 8.0 | 0.37 | - | 100 | 64 | 1.3 |
| Žitkovac | 6 | 23 | 1.9 | 100 | 0.01 | 1.4 | 5.2 | 0.87 | <0.01 | 240 | 200 | 0.10 |
| K. Mitrovica | 6 | 13 | 0.80 | 84 | 0.07 | 1.6 | 1.9 | 0.38 | <0.01 | 97 | 37 | 0.26 |
| Gradac | 3 | 3.2 | 0.08 | 4.2 | 0.03 | 0.22 | <1.0 | 0.19 | <0.01 | 21 | 5.2 | 0.41 |
| Probištip I | 2 | 21 | 0.08 | 6.7 | 0.04 | 13 | 9.5 | 0.68 | <0.01 | 39 | 5.0 | 0.58 |
| Probištip II | 2 | 11 | 0.10 | 4.3 | 0.02 | 6.6 | 7.1 | 0.25 | <0.01 | 22 | 5.0 | 0.13 |
| Probištip III | 2 | 7.7 | 0.04 | 2.4 | 0.02 | 5.4 | 6.1 | 0.30 | <0.01 | 17 | 4.6 | 0.30 |
| Sasa I | 2 | 4.3 | 0.01 | 9.9 | 0.02 | 1.1 | 4.3 | 0.33 | <0.01 | 7.0 | 14 | 0.36 |
| Sasa II | 2 | 5.0 | 0.07 | 16 | 0.02 | 2.0 | 5.7 | 0.33 | <0.01 | 6.0 | 19 | 0.36 |
| Sasa III | 2 | 4.4 | 0.01 | 9.7 | 0.03 | 1.3 | 2.7 | 0.57 | <0.01 | 7.8 | 23 | 0.36 |
| Toranica | 2 | 4.4 | <0.01 | 9.8 | 0.03 | 2.7 | 5.0 | 0.66 | 0.010 | 8.2 | 17 | 0.37 |
| Veles | 3 | 39 | 0.07 | 35 | 0.70 | 67 | 44 | 2.5 | <0.01 | 450 | 23 | 8.3 |
| Lojane | 2 | 0.13 | <0.01 | 0.24 | <0.01 | <0.10 | 100 | <0.10 | 0.01 | 12,000 | 43.6 | <0.10 |
| Bor I | 3 | 2.0 | 0.52 | 1.4 | 0.36 | 0.30 | 46 | <0.10 | <0.01 | 8.5 | 4.8 | <0.10 |
| Bor II | 3 | 1.1 | 0.27 | 1.4 | 0.16 | <0.10 | 7.6 | <0.10 | <0.01 | 2.9 | 5.4 | <0.10 |
| Bor III | 1 | <1.0 | 0.05 | 3.1 | 0.40 | 4.6 | 1100 | 0.12 | 0.11 | 120 | 9.4 | 0.79 |
| Bor IV | - | - | - | - | - | - | - | - | - | - | - | - |
| Bor V | - | - | - | - | - | - | - | - | - | - | - | - |
| Grot | 1 | 1.0 | 0.03 | <1.0 | <0.01 | 0.13 | 5.1 | 0.37 | <0.01 | 4.9 | 19 | 0.27 |
| Lece | 7 | 5.3 | 1.2 | - | 0.04 | 17 | - | 0.14 | - | - | - | 0.55 |
| Rudnica I | 1 | 1.4 | 0.01 | 15 | 0.19 | 7.0 | 6.8 | <0.10 | <0.01 | 4.7 | 71 | <0.10 |
| Rudnica II | - | - | - | - | - | - | - | - | - | - | - | - |
| Rudnik | 1 | 15 | <0.01 | 48 | 0.08 | 0.37 | <1.0 | 0.50 | <0.01 | 16 | 78 | 0.44 |
| Veliki Majdan I | - | - | - | - | - | - | - | - | - | - | - | - |
| Veliki Majdan II | - | - | - | - | - | - | - | - | - | - | - | - |
| Zajača | 1 | 14 | 0.19 | <1.0 | 0.02 | <0.10 | 5.7 | <0.10 | <0.01 | 11,000 | 0.3 | 0.11 |

Data source: 1—surface sampling (RESEERVE), 2—hand drilling/trenching, 3—machine drilling, 4—data from Albanian geological survey, 5—BRGM report [19], 6—surface sampling/archive material–Ristović, I., 7—personal archive (I. Ristović), 8—data from the article by Grieco et al. [42]. Important values are shown in red.

Samples (surface sampling, hand drilling, and machine drilling) collected at the selected sampling sites were taken to the laboratory where a pre-analytical sample preparation was carried out. All samples were dried in an oven at 40 °C, followed by crushed in a mortar, and sieved under the reach grains less than 125 μm. These samples were prepared for chemical analysis. The elemental contents of 65 elements were determined by ICP–MS at Bureau Veritas Commodities Canada Ltd.(Richmond, BC, Canada) [43]; accredited under ISO 9001:2015) after aqua regia digestion (at 95 °C, using the 1DX method) and total 4-acid digestion following international standards (ISO 14869-1:2001, 2001). However, the study presents only elements that are most promising for future recovery (Ag, Au, Bi, Cu, In, Mo, Pb, Re, Sb, W, and Zn).

**Table 3.** Basic data of potentially prospective SRM deposits.

| Location | Remark | Ag (t) | Au (t) | Bi (t) | Cu (Kt) | In (t) | Mo (Kt) | Pb (Kt) | Re (t) | Sb (Kt) | W (Kt) | Zn (Kt) |
|---|---|---|---|---|---|---|---|---|---|---|---|---|
| Fushë Arrëz | 8 | - | 2.4 | - | 5.9 | - | - | - | - | - | - | - |
| Kukës | 4 | - | - | - | 5.1 | - | - | - | - | - | - | - |
| Rehove | 4 | - | - | - | <1.0 | - | - | - | - | - | - | - |
| Reps | 4 | - | - | - | 6.7 | - | - | - | - | - | - | - |
| Rreshen | 4 | - | - | - | <1.0 | - | - | - | - | - | - | - |
| Sase I | 1 | - | - | - | - | - | - | - | - | - | - | - |
| Sase II | 1 | - | - | - | - | - | - | - | - | - | - | 5.1 |
| Veovača | 1 | 65 | - | - | - | - | - | - | - | - | - | - |
| Badovac | 6 | 120 | 4.6 | - | - | - | - | 47 | - | - | - | - |
| Kišnica | 5 | 52 | 4.7 | 100 | - | - | - | 12 | - | - | - | 27 |
| Leposaviæ I | 6 | - | - | - | - | - | - | - | - | - | - | - |
| Leposaviæ II | 6 | - | - | - | - | - | - | - | - | - | - | - |
| Novo Brdo | 5 | 10 | 1.3 | - | - | - | - | 2.6 | - | - | - | 5.3 |
| Gornje polje I | 6 | - | 8.5 | - | - | - | - | - | - | - | - | - |
| Gornje polje II | 6 | 140 | 9.0 | 710 | - | - | - | 66 | - | - | 1.3 | - |
| Gornje Polje III | 6 | 220 | - | 190 | 6.7 | - | - | 130 | - | 2.7 | - | 200 |
| Žarkov potok | 5 | 100 | 2.1 | 600 | - | - | - | - | - | - | - | 130 |
| Žitkovac | 6 | 270 | 23 | 1200 | - | - | - | 100 | - | - | 2.4 | - |
| K. Mitrovica | 6 | 40 | 2.4 | 250 | - | - | - | - | - | - | - | - |
| Gradac | 3 | - | - | - | - | - | - | - | - | - | - | - |
| Probištip I | 2 | 150 | - | - | - | 89 | - | 48 | - | - | - | 41 |
| Probištip II | 2 | 130 | 1.2 | - | - | - | - | - | - | - | - | - |
| Probištip III | 2 | - | - | - | - | - | - | - | - | - | - | - |
| Sasa I | 2 | - | - | - | - | - | - | - | - | - | - | - |
| Sasa II | 2 | - | - | - | - | - | - | - | - | - | - | - |
| Sasa III | 2 | - | - | - | - | - | - | 57 | - | - | - | - |
| Toranica | 2 | - | - | - | - | - | - | 66 | - | - | - | - |
| Veles | 3 | 71 | - | - | 13 | 120 | - | 45 | - | - | - | 150 |
| Lojane | 2 | - | - | - | - | - | - | - | - | 6.0 | - | - |
| Bor I | 3 | - | 4.2 | - | 29 | - | - | - | - | - | - | - |
| Bor II | 3 | - | 6.0 | - | 36 | - | - | - | - | - | - | - |
| Bor III | 1 | - | - | - | 200 | - | 55 | - | 5.3 | - | - | 390 |
| Bor IV | - | - | - | - | - | - | - | - | - | - | - | - |
| Bor V | - | - | - | - | - | - | - | - | - | - | - | - |
| Grot | 1 | - | - | - | - | - | - | - | - | - | - | - |
| Lece | 7 | - | 3.3 | - | - | 46 | - | - | - | - | - | 15 |
| Rudnica I | 1 | - | - | - | 5.8 | - | - | - | - | - | - | - |
| Rudnica II | - | - | - | - | - | - | - | - | - | - | - | - |
| Rudnik | 1 | 130 | - | - | - | - | - | 44 | - | - | - | - |
| Veliki Majdan I | - | - | - | - | - | - | - | - | - | - | - | - |
| Veliki Majdan II | - | - | - | - | - | - | - | - | - | - | - | - |
| Zajača | 1 | 8.5 | 0.1 | - | - | - | - | - | - | 6.6 | - | - |
| **WB** | | **1500** | **73** | **3100** | **310** | **260** | **55** | **620** | **5.3** | **15** | **3.7** | **970** |

Data source: 1—surface sampling (RESEERVE), 2—hand drilling/trenching, 3—machine drilling, 4—data from Albanian geological survey, 5—BRGM report [19], 6—surface sampling/archive material (I. Ristović), 7—personal archive (I. Ristović), 8—data from the article by Grieco et al. [42]. Kt-kilotons.

The quality control was provided by duplicates and certified reference samples (DS8, OREAS 24P, and OREAS45CA). For selected elements, it was found that precision for Ag was 13.6, Au 17, Cu 4.4, Mo 7.3, Pb 2.8, Sb 5.2, and Zn 4, and accuracy for Ag was 1.6, Au 1.5, Cu 1.3, Mo 3.6, Pb 3.3, Sb 1.3, and Zn 2.0.

## 3. Results

### 3.1. Overview of Secondary Raw Deposits in the West Balkan Region

The results presented in the study by Šajn et al. [28] are valorized and disseminated in this article. In total, there are 1650 waste disposal sites in the West Balkan region. Of

these, 1540 (93%) belong to mining waste landfills, 67 (4%) to processing waste landfills, and 43 (2%) to metallurgical waste landfills. In total, they cover an area of about 65 km$^2$, of which 36 km$^2$ (55%) are landfills for mining waste, 24 km$^2$ or 37% are landfills for processing waste, and 5 km$^2$ (8%) are landfills for metallurgical waste; 1460 sites were presented in the previous study [28] describing the process of formation of the mineral register within the project RESEERVE in the West Balkan, while 190 sites are new, mostly located on the territory of Kosovo *.

The area covered with mining, processing, and metallurgical slag heaps in Kosovo is 260, 220, and 70 ha, respectively. However, most of the selected SRM deposits are in Serbia, where 1040, 1200, and 110 ha are covered with mining, processing, and metallurgical slag landfills, respectively. On the other hand, Croatia has the smallest area covered with these three types of waste landfills: 140, 31, and 37 ha for mining, processing, and metallurgical slag landfills, respectively. Therefore, the waste landfills in Croatia were excluded from further analysis.

The main elements in the deposits studied are Al, Cr, Cu, Fe, Ni, Mn, Pb, Sb, and Zn, with a variety of trace elements, e.g., Ag, Au, Bi, and W. Copper mining generally generates most mining waste dumps. The largest amount of such waste is located in Serbia and covers more than 980 ha, followed by 520 ha generated from mining of Fe–Mn deposits in Bosnia and Herzegovina. As far as mining wastes are concerned, copper flotation residues make up the largest part of the landfills. In Serbia, these wastes cover an area of 1050 ha. Of the metallurgical slag landfills, the largest area (68 ha) is in Bosnia and Herzegovina and is part of the Fe–Mn smelting plant. The smallest amount of waste, on the other hand, usually comes from Sb mining and metallurgy.

### 3.2. Potentially Prospective SRM Deposits

According to the criteria for determining prospective landfills (the size of tailings should be larger than 0.5 million tons; it needs to have a promising chemical composition and to be in an inactive, abandoned or nonremediated state, where older flotation methods were used and where the possibility of recovery methods exist), data collected, and additional chemical analyses, and in accordance with zero-waste philosophy, 42 promising landfills could be selected, where around 270 million tons of material is stored and covers an area of 750 ha (Table 1, Figure 3). These landfills are mostly processing waste landfills (35), while 7 landfills belong to metallurgical waste landfills; of these prospective SRM deposits, 29 waste landfills or 145 million tons of material are identified as abandoned landfills, while the remaining ones are in an active state. Currently, these deposits store around 120 million tons of material; however, their quantity continues to grow.

Most prospective SRM landfills are in Serbia and Kosovo * (13 and 11, respectively). The locations in Kosovo * that are the richest with valuable metals are shown in green. In addition, 9 belong to North Macedonia, 5 to Albania, while Bosnia and Herzegovina and Montenegro have 3 and 1 waste landfills, respectively. Most of these materials originate from Pb–Zn mining and metallurgical activities. Almost 160 million tons of material is stored in an area of 540 ha, followed by Cu mining activities, where there are 200 ha with 110 million tons of material. The remaining material comes from Sb mining: 7 ha and 1.1 million tons of material.

Table 2 shows the chemical composition of samples collected on secondary raw deposits in Albania, Bosnia and Herzegovina, Kosovo *, Montenegro, North Macedonia, and Serbia. The values that exceed the following predefined values are indicated in red: Ag > 10 g/t, Au > 0.10 g/t, Bi > 50 g/t, Cu > 0.10%, In > 10 g/t, Mo > 1000 g/t, Pb > 0.50%, Re > 0.10 g/t, Sb > 1000 g/t, W > 100 g/t, and Zn > 0.50%. The highest Ag concentration, as much as 86 g/t, is characterized for Gornje Polje III in Zvečan, Kosovska Mitrovica. This location has also the highest concentrations of Pb, as much as 5.1%. High Au concentrations, 2.5 and 2.3 g/t, were determined in Novo Brdo and Kišnica (Gračanica), both situated in Kosovo *. Žitkovac, as a part of Trepča mine district, has the highest concentration of Bi (100 g/t) and W (200 g/t). Veles, North Macedonia, has the highest values of Cu

(0.70%), In (67 g/t), and Zn (8.3%). In the Bor mining district, the highest content of Mo and Re, 1100 g/t and 0.11 g/t, respectively, was detected. Lojane has the largest content of Sb (12,000 g/t). High Sb content was also determined in Zajača (11,000 g/t).

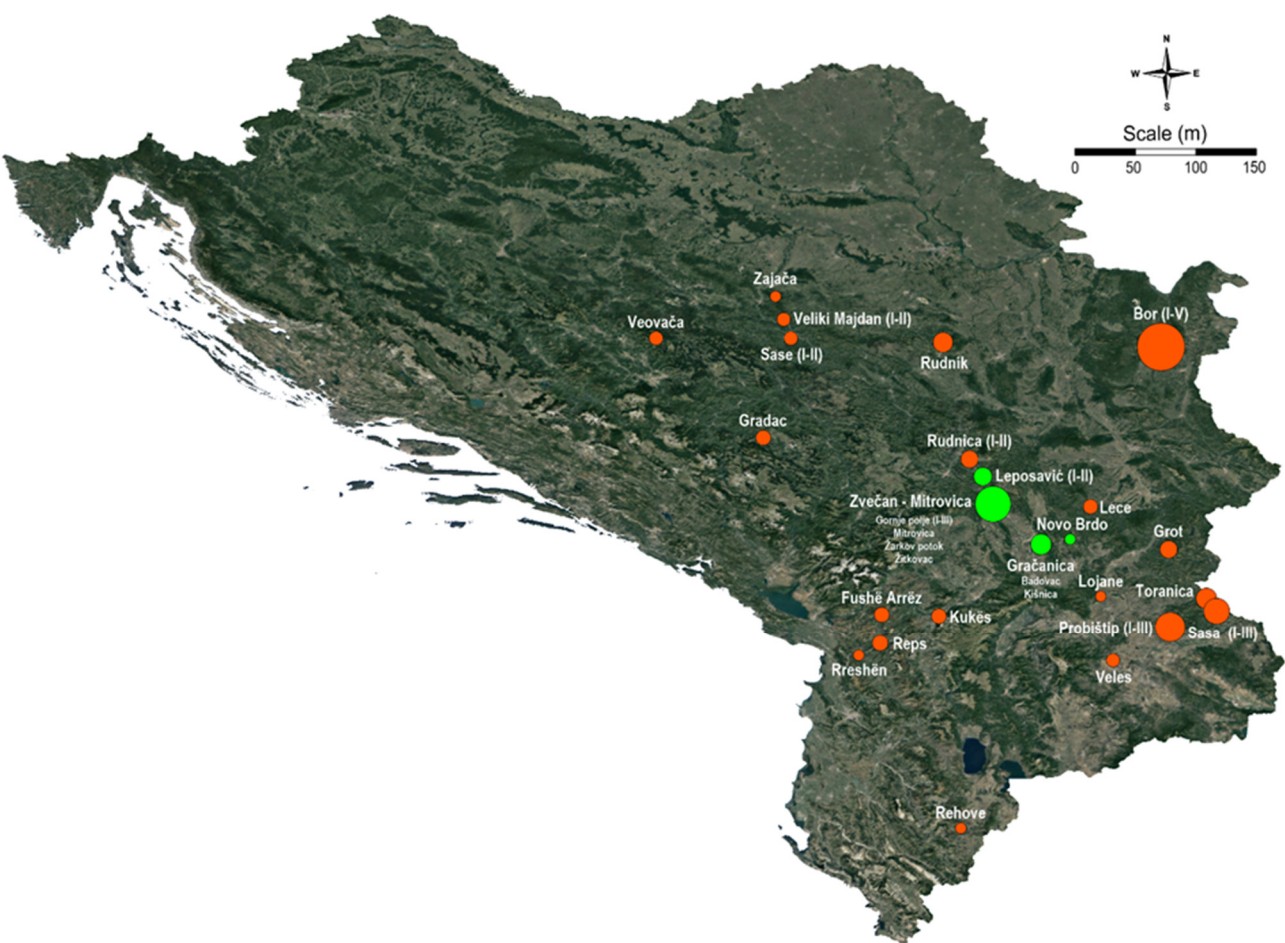

**Figure 3.** Locations of potentially prospective landfills (the size of the circle is proportional to the size of the SRM deposit).

Table 3 shows the set of possible valuable elements. Žitkovac, with its 270 tons, contains the largest amount of Ag. There are also the highest amounts of Au (23 tons) and Bi (1200 tons). The tailings at Bor III contain the largest amount of Cu, 200,000 tons, and significant amounts of Mo, Re, and Zn: 55,000, 5.3, and 390,000 tons, respectively. The largest amount of Pb (130,000 tons) was found in the Gornje Polje III tailings. In addition to these elements, some tailings are also rich in In, Sb, and W. For example, the tailings from Veles contain up to 120 tons of In, while the tailings from Zajača contain up to 6600 tons of Sb. In two deposits, namely in Gornje polje II and Žitkovac, the presence of W was detected, 1300 tons and 2400 tons, respectively. In total, the investigated deposits contain 1500 tons of Ag, 73 tons of Au, 3100 tons of Bi, 310,000 tons of Cu, 260 tons of In, 55,000 tons of Mo, 620,000 tons of Pb, 5.3 tons of Re, 15,000 tons of Sb, 3700 tons of W, and 970,000 tons of Zn.

Silver and gold, together with the platinum group metals, belong to the so-called precious metals. The annual production in 2021 was 3030 tons of gold and 24,000 tons of silver [44]. Gold production has generally increased since 2005, while silver production has remained the same with minor fluctuations. Silver has the highest electrical and thermal conductivity of all metals and is therefore used in the production of printed electrical circuits and as a vapor-deposited coating for electronic conductors. Unlike gold, which usually occurs in native form, silver is widely distributed in many naturally occurring

minerals (e.g., tetrahedrite, argentite). Both elements can also occur in small proportions in lead and copper ores as a byproduct, adding value to these deposits [45].

In addition to silver and gold, other elements are also abundant in the study area. Zinc and copper are essential elements needed for the well-being of plants, animals, and humans. Copper occurs as native copper and in mineral form (e.g., $CuFeS_2$, $Cu_5FeS_4$, $Cu_2CO_3(OH)_2$), while minerals rich in zinc are sphalerite (ZnS), smithsonite ($ZnCO_3$), zincite (ZnO), and ganite ($ZnAl_2O_3$). Both elements also occur in many minerals such as olivine, magnetite, pyroxene, amphibole, and biotite [46]. The annual production of these two elements has increased since 2005. In 2021, the global production was 12.8 million tons of zinc and 21 million tons of copper [44]. The main anthropogenic sources of Zn and Cu are mining and metallurgical activities. Zinc is also used in the automotive industry, in alloys and electroplating processes, and in the paint industry, while copper is found in industrial dust, industrial wastes, in the use of copper chemicals, and elsewhere [46].

The most common metal produced as a byproduct in the processing of copper and zinc is bismuth. The production of this metal has steadily increased yearly. In 2020, 16,000 tons of bismuth was produced in refineries worldwide [44]. Bismuth is used in various fields such as industry, laboratory, cosmetics, and pharmaceuticals. Owing to its low toxicity, it is considered a green metal for the environment and is usually used as a substitute for lead in plumbing fixtures, free-machining steels, and as a metallurgical additive in foundries. It is also widely used in medicine and healthcare [47].

Tungsten and molybdenum have some similar basic properties and are considered refractory metals due to their corrosion resistance, excellent conductivity, and extremely high melting point. Therefore, they are available for plating products. Molybdenum is also used for the production of alloys, especially steel alloys to increase hardness, strength and electrical conductivity, while the applications of tungsten range from the household needs to high-tech special applications [48,49]. Tungsten production has been constant with minor variations in certain years. In 2020, the annual production was 84,000 tons while molybdenum production has generally increased since 2015 and reached 300,000 tons in 2020 [44].

Another metal with a high melting point is rhenium. It is mainly extracted from the host ore molybdenite in porphyry copper deposits [50]. The global mine production of rhenium has been estimated to be 50 to 60 tons annually [44]. It is commonly used as an additive in the production of superalloys along with iron, cobalt, nickel, tungsten, and molybdenum, while rhenium compounds are used as catalysts in industries such as petrochemical, pharmaceutical, and organic synthesis processes [51].

## 4. Discussion

### 4.1. Locations and Metals for Potential Recovery

After a thorough review, it was concluded that the elements with the highest potential for further recovery using modern and efficient technology are the following: Ag, Au, Cu, and Zn (Figures 4 and 5). As far as the metal content is concerned, Ag predominates in the tailings of Gornje Polje, but when the amount of metal is considered, it predominates in the tailings of Žitkovac, followed by Gornje Polje. As for silver, it is worth mentioning that the highest amount of gold is found in the Žitkovac tailings. However, the highest concentrations were measured in the Novo Brdo tailings. All the aforementioned deposits, i.e., Gornje Polje, Žitkovac, and Novo Brdo are located in Kosovo * and belong to either the Novo Brdo mining district or the Stari Trg mine. Both mining districts are rich in lead, zinc, and silver [18]. The content and quantity of copper and zinc appear similar. The concentrations of the two elements are highest in Veles (North Macedonia), while the amounts of these two elements predominate in the tailings of Bor (Serbia). On the other hand, the measurements in this tailings pile showed that the Zn content is less than 1%, which is much lower than the values determined in Veles, 8.5%. It should be mentioned that the Bor mine is a porphyritic deposit. This type of deposit in the Balkan region has a relatively low copper content but is profitable due to its size [39], while in

Veles, metallurgical slags are stored, which generally have higher concentrations of metals than processing and mining waste.

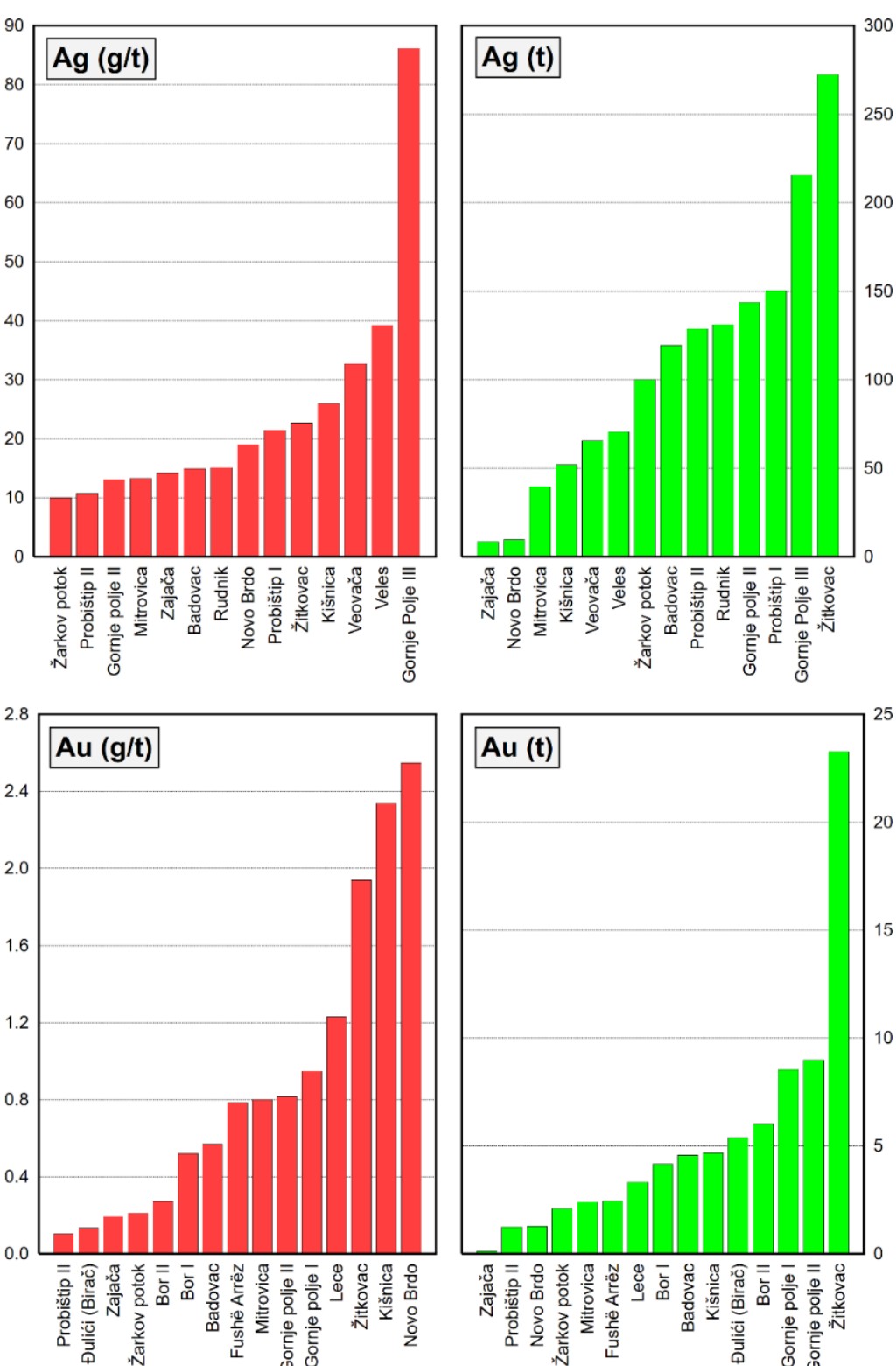

**Figure 4.** Content and quantity of silver and gold in prospective landfills.

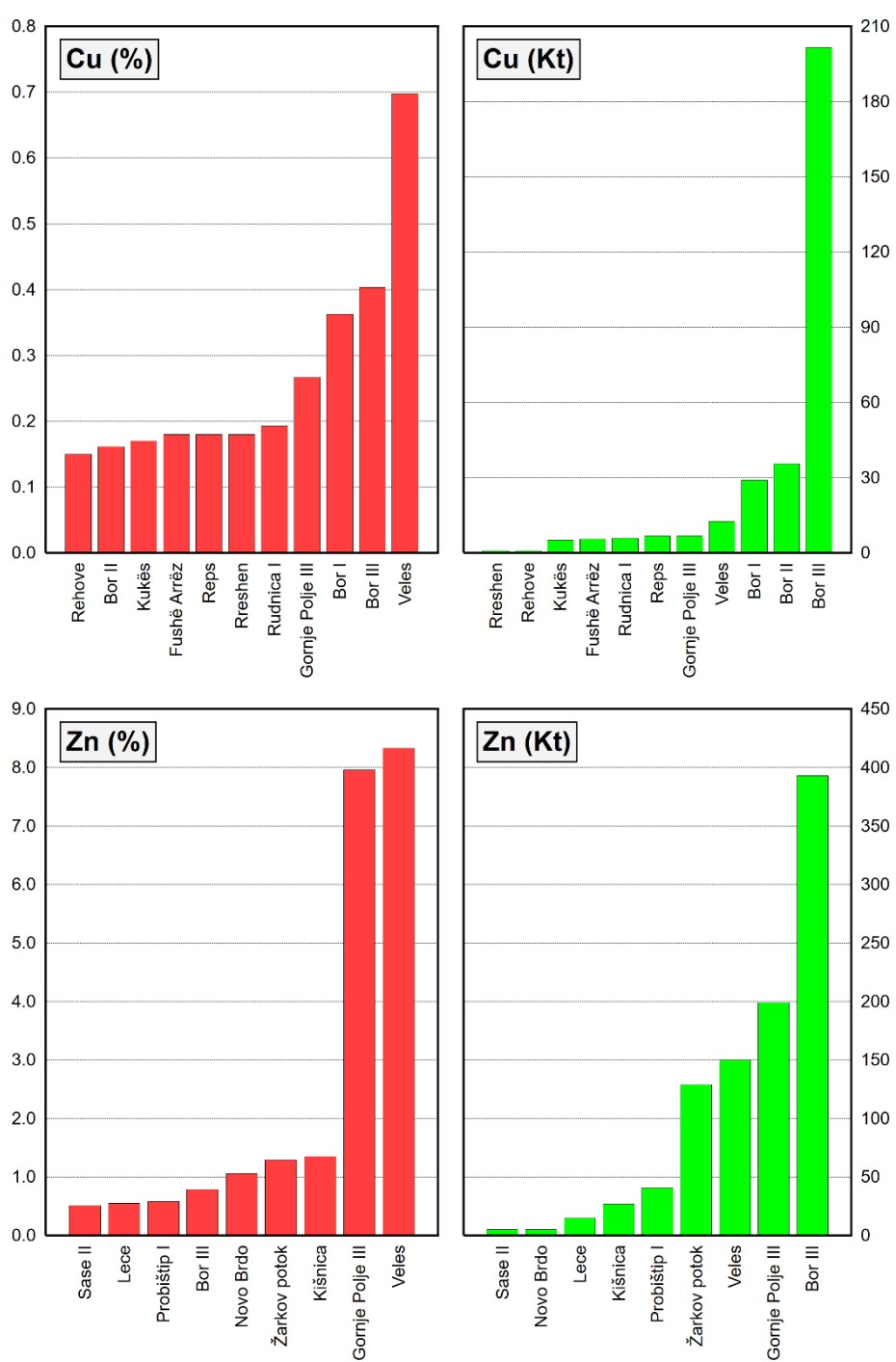

**Figure 5.** Contents and quantities of Cu and Zn in prospective landfills.

The Bor mine area and the Trepča mine area seem to be the most promising sites for further recovery of elements (Figure 6). The Bor porphyry copper deposit has five sampling sites (Bor I, Bor II, Bor III, Bor IV, and Bor V), covering an area of 154 ha and containing 97 million tons of material. In terms of valuable elements, it consists of 10 tons of Au, 270,000 tons of Cu, 55,000 tons of Mo, 5.3 tons of Re, and 390,000 tons of Zn suitable for mining. Molybdenum and rhenium are characteristic elements in porphyry copper deposits. In fact, Mo forms an important ore component in porphyritic Cu deposits, while rhenium occurs mainly in molybdenite, in which ReS$_2$ forms a solid solution with MoS$_2$ [50]. In addition to Mo and Re, porphyritic copper deposits are also sources of gold.

Gold occurrences in this type of deposit vary widely and were originally determined based on the Au/Mo ratio [52].

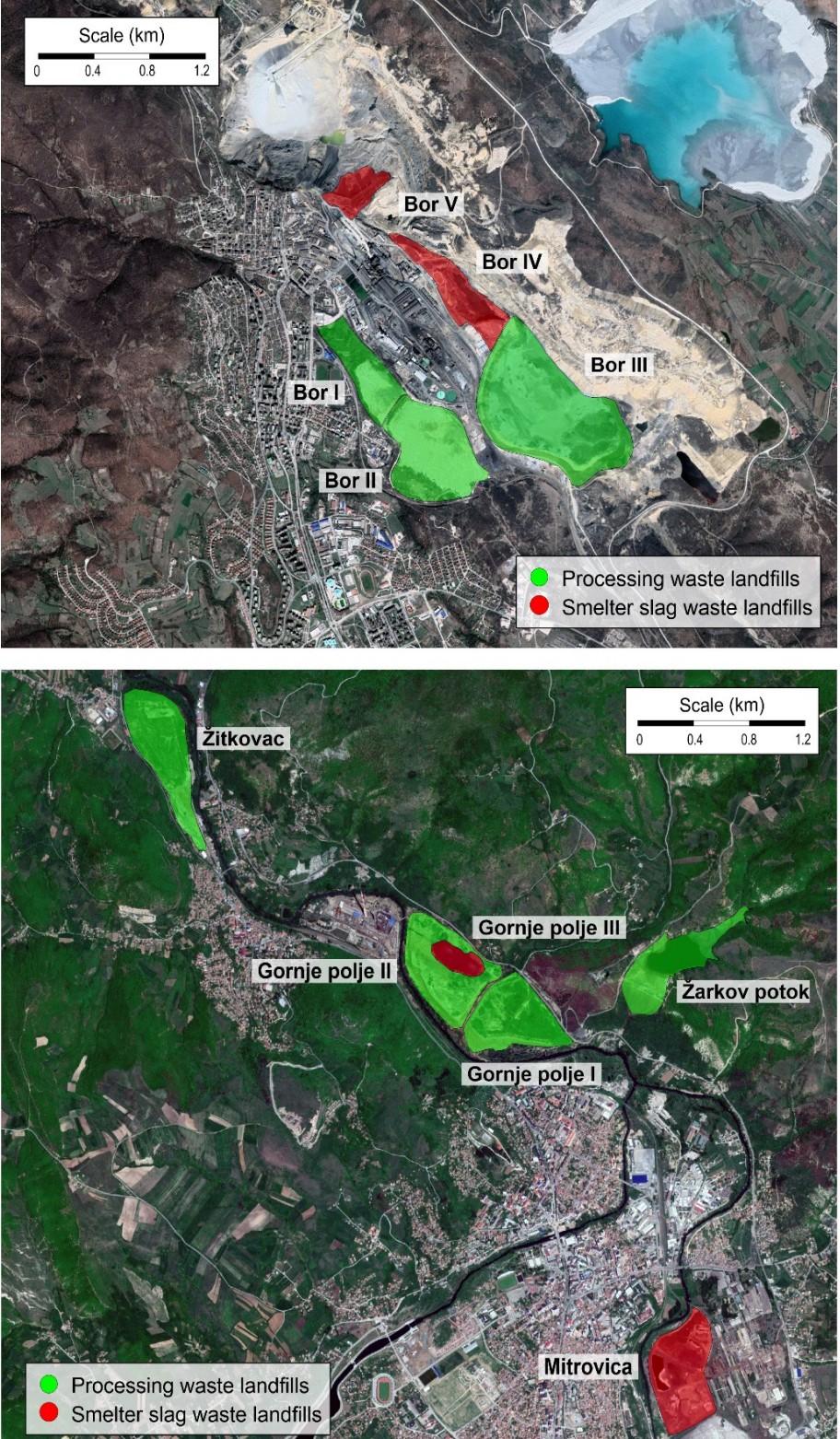

**Figure 6.** Bor (**above**) and Trepča (**below**) mine districts.

The second promising deposit is the Trepča mine area (Figure 6). It is classified as a carbonate-replacement deposit [53], but several other types of Pb–Zn mineralization can be

detected nearby (hydrothermal, skarn, veins, etc.) [19]. It is the best-known Zn–Pb (Ag) mine in the Balkans and contains several trace elements in addition to the main elements. The presence of gold can be considered as an expression of the same mineralization phenomenon related to Oligocene and Miocene volcanism [19]. Antimony is characteristic of hydrothermal Pb–Zn deposits and forms mainly sulfides or composite sulfides, either with Hg, As, Tl, Ag, Au, Bi, or Pb [54]. Copper is associated with chalcopyrite, tetrahedrite, etc.; the occurrence of bismuth is associated with galena, while scheelite contains tungsten [19]. After detailed investigations, it was concluded that the Trepča mining district, considering its five waste dumps (Žitkovac, Gornje Polje I, Gornje Polje II, Gornje Polje III, Žarkov Potok, and Kosovska Mitrovica) with an area of 144 ha and 48 million tons of material, contains the following quantities of metals: 770 tons Ag, 45 tons Au, 3000 tons Bi, 6700 tons Cu, 300,000 tons Pb, 2700 tons Sb, 3700 tons W, and 330,000 tons Zn.

### 4.2. Economical Potential of Historical Waste Landfills

Table 4 shows metal prices in three different time periods: 2000; 2020, shortly after the start of the pandemic COVID-19; and metal prices that were on the market in March 2022, reflecting current political conditions. The prices for Ag, Au, Cu, Mo, Pb, and Zn were obtained from the website Trading Economics [55], more precisely on 28 March 2000, 28 March 2020, and 28 March 2022, respectively. Prices for the remaining elements were obtained by applying the following websites: Shanghai Metals Market (SMM) [56], United States Geological Survey (USGS) [57], and Metalary [58], considering the average metal price for March 2000, 2020, and 2022, respectively.

**Table 4.** Metal rates and values for selected chemical elements according to the years 2000, 2020, and 2022.

| Element | Metal (t) | Rate (USD/kg) | | | Millions USD | | |
|---|---|---|---|---|---|---|---|
| | | **2000** | **2020** | **2022** | **2000** | **2020** | **2022** |
| Ag [1] | 1500 | 192 | 463 | 884 | 289 | 698 | 1330 |
| Au [1] | 73 | 10,300 | 53,900 | 68,600 | 752 | 3940 | 5010 |
| Bi [2] | 3100 | 6.0 | 7.72 | 7.17 | 18 | 24 | 22 |
| Cu [1] | 310,000 | 1.27 | 4.81 | 10.5 | 394 | 1490 | 3260 |
| In [2] | 260 | 188 | 220 | 299 | 48 | 56 | 76 |
| Mo [1] | 55,000 | 5.90 | 24.8 | 45.8 | 327 | 1370 | 2540 |
| Pb [1] | 620,000 | 0.387 | 1.57 | 2.38 | 240 | 972 | 1470 |
| Re [2] | 5.3 | 1110 | 1030 | 3550 | 6 | 5 | 19 |
| Sb [2] | 15,000 | 1.50 | 5.87 | 12.8 | 23 | 90 | 196 |
| W [2] | 3700 | 8.20 | 30.3 | 49.3 | 30 | 113 | 183 |
| Zn [1] | 970,000 | 1.08 | 1.91 | 4.10 | 1040 | 1840 | 3950 |
| **SUM** | | | | | **3170** | **10,600** | **18,100** |

Legend: [1] data from [55]; [2] data from [56–58].

Gold reached the highest price in all three years. The price increased fivefold from 2000 to 2020 and almost sevenfold in March this year compared to 2000. A high price difference is also observed in silver. The price doubled from 2000 to 2020, while the price for silver in 2022 is double the price during 2020. The price for copper almost quadrupled in 2020 compared to the price during 2000, and then increased by two orders of magnitude from 2020 to 2022. However, an even larger increase can be seen in the price of molybdenum. The price quadrupled from 2000 to 2020, while the price in 2022 is double the price in 2020. Other elements such as Pb, Zn, In, Sb, and W also have a higher price today than two years ago and before. These prices are on average twice as high today as they were in 2020, but in the case of Bi, higher prices were observed in 2020 than in 2022, while in the case of Re, higher prices were observed in 2000 compared to 2022 prices. However, the highest Re rate is observed for 2022.

In accordance with the March 2020 metal prices, it is estimated that the recovery of tailings in the West Balkan could bring up to 10,600 million USD, while in March 2022 this figure is estimated at 18,100 million USD. However, if we examine what the same quantities of elements would have brought in 2000, there would have been a much smaller profit—3170 million USD. For the same quantities of mine waste, under today's circumstances, the profit is almost six times higher than in 2000 and almost two times higher than in 2020.

Table 5 shows the potential profit from the recovery of metals from mine waste, considering the countries of the West Balkan. It is estimated that the highest profit from mining tailings can be obtained in Serbia, followed by Kosovo *. The tailings from Serbia are mostly rich with Cu, Mo, and Zn, and their recovery can bring up to 8340 million USD profit, while the recovery of the tailings of Kosovo * can bring 7310 million USD profit. The tailings of Kosovo * are rich mainly with Au, Ag, Pb and Zn. In addition to Kosovo * and Serbia, Northern Macedonia also has high potential for mining other metals. Mining these tailings, where Pb and Zn have the greatest potential, can yield a profit of 1960 million USD. Similar to Northern Macedonia, Kosovo *, and Serbia, both Albania and Bosnia and Herzegovina are also countries with high potential for further mining. The profit is estimated at 369 million and 448 million USD for Albania and Bosnia and Herzegovina, respectively.

**Table 5.** Metal values per country (March 2022) in million USD.

| Element | Albania | B and H | N. Macedonia | Kosovo * | Serbia |
|---|---|---|---|---|---|
| Ag | - | 58 | 309 | 842 | 124 |
| Au | 168 | 369 | 85 | 3830 | 934 |
| Bi | - | - | - | 22 | - |
| Cu | 201 | - | 132 | 70 | 2850 |
| In | - | - | 62 | - | 14 |
| Mo | - | - | - | - | 2540 |
| Pb | - | - | 514 | 856 | 104 |
| Re | - | - | - | - | 19 |
| Sb | - | - | 77 | 35 | 84 |
| W | - | - | - | 183 | - |
| Zn | - | 21 | 781 | 1480 | 1670 |
| **SUM** | **369** | **448** | **1960** | **7310** | **8340** |

Examining the quantities of valuable elements stored in the West Balkan tailings, Zn is the most promising, followed by Pb, Cu, and Mo (Figure 7a). Indeed, the countries of the West Balkan are particularly rich in Pb–Zn ore deposits and Cu porphyry deposits. However, if we look at metal prices, gold has the highest value, which is often found in different types of ore deposits, depending on the mineralization process. Besides Au, the metals Re, Ag, and In are also highly traded in the market. In terms of elements to be recovered, gold is the most promising (Figure 7b,c). The total recovery is estimated at about 5010 million USD, followed by Zn, Cu, and Mo with about 3950, 3260, and 2540 million USD profit, respectively. In addition to Au, Zn, Cu, and Mo, both Pb and Ag are also promising prospects for mining. Mining of Pb and Ag can yield up to 1470 million and 1330 million USD in profit, respectively. Sb, W, In, Bi, and Re, on the other hand, have much lower potential for mining, despite their high market price, and are limited to certain locations in the West Balkan.

It is apparent that because of the huge number of tailings and the desire to protect the environment and prevent failures, the future production of valuable raw materials will be based on the exploitation of tailings and lower grade ore [59]. Several studies have examined the prospect of metal recovery from secondary deposits. One study [60] focused on the abandoned tailings in Chile, Mexico, and Australia, and concluded that in Mexico, the most prospective elements are Bi, Sb, W, In, Mo, and Zn, while Chile has significant potential for Bi, Sb, W, Mo, and Zn. The data from Australia are more complicated, however,

owing to several factors (i.e., different ore processing between areas, climates). For example, Northwest Queensland has the potential for Co. The prospect of recovery depends on the development of technology [61], on the one hand, and metal prices on the other. Namely, the metal prices can be strongly diverse. For instance, in the period between 2000 and 2022, the price of gold ranged between 10,300 and 68,600 USD/kg; similar changes can be also observed for silver (192 to 884 USD/kg), copper (1.27 to 10.5 USD/kg), and tungsten (8.2 to 49.3 USD/kg) (Table 6). The study by [62] concluded that according to the prices from 2014, the potential value of Ag and Cu from landfills, where the tonnage was estimated at approximately 34 tons and 19,200 tons, was 25 million and 134 million USD, respectively. While one study [8] determined that by applying the proper technique, 696 mine tailings (mainly copper mining tailings) from Chile where 46,110 tons of vanadium is stored, can bring 76 million USD of profit.

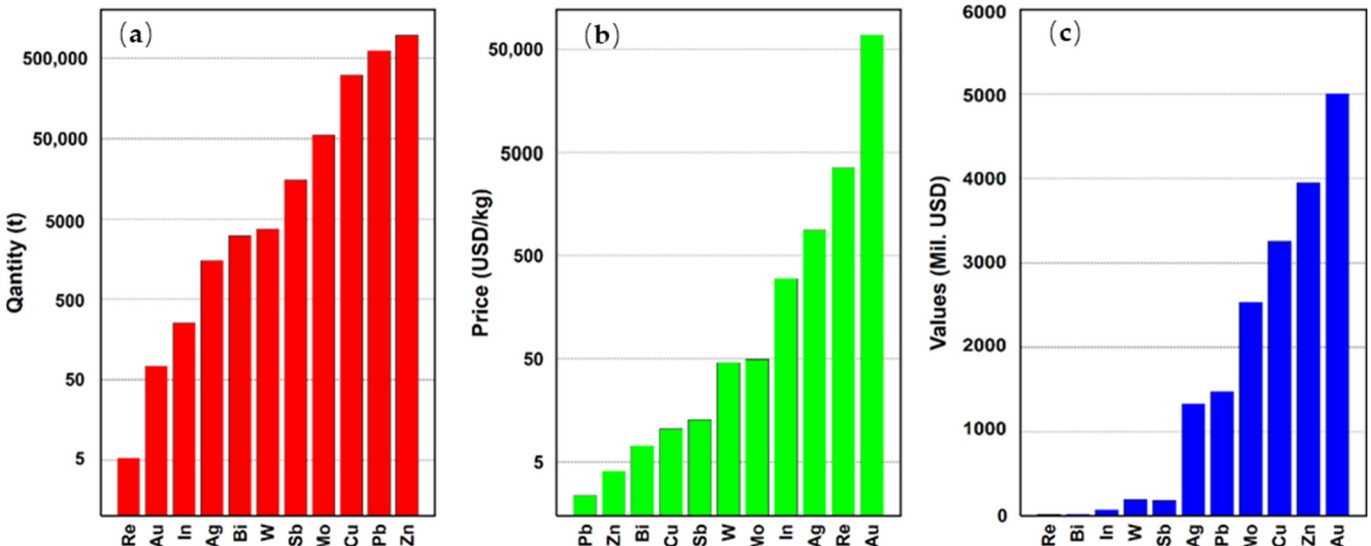

**Figure 7.** Quantity (**a**), prices (**b**), and values (**c**) of valuable elements.

### 4.3. Recovery Methods

Considering the total quantities of secondary mineral commodities to which access is possible, it can be found that they are huge, measured in millions of tons (Table 3). Considering their composition, it should be particularly emphasized that, in addition to useful elements (metals), they also contain some valuable elements, e.g., Au and Ag, whose demand on the European market is very high, along with other elements. Considering the fact that the EU imports a large number of critical elements, mainly from abroad [1], the question arises why these useful secondary raw materials are not extracted from the tailings and dumps of both the active mines and closed and abandoned mines in southeastern Europe, of which there are many.

One of the main reasons for the low utilization of these resources is certainly the insufficiently developed methodology for the recovery of useful elements from tailings and dumps. Indeed, if we look at all the scientific literature that can be found in the countries of the West Balkan and worldwide, we find that the methods of extracting and recycling valuable elements from tailings are mainly limited to experimental trials and research in laboratories. Although the recovery of useful elements from mine tailings is being tested worldwide, the results vary drastically from experiment to experiment. These differences are related to the use of different recovery methods, the composition of the mine waste, and the methods of preparation and processing of the mineral raw materials, which vary from mine to mine.

The characterization and elemental composition of the useful elements determined in mining wastes also differ from case to case, since they are usually extracted by different

methods, sometimes by shallow drilling and sometimes by deep drilling. Therefore, the question of the representativeness of the samples also arises in this part, considering that the mining wastes were deposited in the deep parts of the tailings about 100 years ago, depending on the beginning of the exploitation of the primary mineral resources. These mining wastes are certainly richer in useful elements than the part of the mining wastes in the higher layers of the tailings because the technologies for preparation and processing of mineral resources were certainly simpler and less effective 100 years ago than they are today.

Overall, knowledge of recovery methods is still based on laboratory results, with little information available on recovery methods in semi-industrial and industrial plants. The objective of this article is therefore to highlight the number of valuable elements in mine tailings, while further research will certainly address applicable and economically viable methods of extracting useful elements. In any case, the economics of recovery, and the legal regulations in this area, have not yet been adequately researched and still vary from case to case, or more precisely, from country to country. The development of technologies to recover metals from industrial waste is a new area of exploitation and recovery of useful elements. It is expected that a greater amount of research will be conducted in the near future in the area of the economics and cost effectiveness of this type of mining.

The study by Krishnan et al. [63] describes in detail the metallurgical recovery of solid wastes using pyrometallurgical technique and hydrometallurgical technique. These methods certainly provide good results in the recovery of useful elements, but the mechanism of using pyrometallurgical technique and hydrometallurgical technique is questionable because they are "dirty methods" that require much energy, and environmental regulations are becoming stricter worldwide. The study proves the following results: (i) 98% and 99% of the gold was recovered from alluvial components; (ii) 8.8 and 230 tons of Hg from Pb and Zn smelters can be recovered by Hg reduction technologies applied to flue gases; and (iii) recovery of metals from electroplating slimes such as gold (Au), silver (Ag), copper (Cu), and zinc (Zn) by combined hydro- and pyrometallurgical processes yields maximum recoveries of 80% Ag, 73% Zn, and 63% Cu, etc. Similar studies, in which the authors examined different methods for the possible recovery of metals from mining waste, are listed in Table 6.

For example by applying chloride leaching in flotation tailings, the final recovery increased up to 98% for Cu, while these values were 93%, 83%, 76%, and 80% for Ni, Zn, Co, and Fe [64]. By applying the carbothermic reduction method, the final recovery increased up to 90% for V and 95% for Cr from tailings from the vanadium extraction process [65]. The study by Syed et al. [66] concluded that there are several methods for gold recovery from secondary sources by applying environmentally friendly technologies, more precisely varieties of leaching, cementing, reducing agents, peeling, coagulants, adsorbents, agglomeration solvents, ion exchange resins, and biosorbents. Furthermore, a study by Rossini and Bernardes found that copper, zinc, and nickel can be recovered from coal wastes of sulfating roasting method [67]. The study by Gunarathne et al. [68] describes the hydrometallurgical approaches for metal recovery from various types of materials. It concluded that future studies should focus on development of environmentally friendly methods of metal dissolution. One of the techniques is also bioleaching, which application is also described in study by Krebs [69], while another study focused on diverse bio-hydrometallurgical processes that can be used for the recovery of nonferrous metals from mining waste [70]. Currently, the need for REE is also growing. The study by Peiravi et al. [71] examines the extraction methods of these elements from coal byproducts, iron ore tailings, apatite, and phosphate byproducts.

**Table 6.** An overview of some researched methodologies for extraction of useful elements from mining waste.

| | Method(s) | Metals | Results | Ref. |
|---|---|---|---|---|
| 1 | chloride leaching | gold, copper, cobalt, nickel, and zinc | final extraction increased up to Cu (98%), Ni (93%), Zn (83%), Co (76%), and Fe (80%). | [64] |
| 2 | carbothermic reduction method | tailings from the vanadium extraction process | final extraction increased up to V (90%) and Cr (95%) | [65] |
| 3 | varieties of leaching, cementing, reducing agents, peeling, coagulants, adsorbents, agglomeration solvents, ion exchange resins, and biosorbents | gold from secondary sources | gold | [66] |
| 4 | - | copper, zinc and nickel and the sulfating agent used was pyrite, from coal wastes | Cu, Zn, Ni | [67] |
| 5 | pyrometallurgical, electrometallurgical, hydrometallurgical bioleaching (chemical leaching, acid leaching, alkaline leaching, thiosulfate leaching, thiourea leaching, halide leaching, cyanide leaching | industrial sludge | Ni, Cu, Zn, Cr, Mn, Fe | [68] |
| 6 | review methods | industrials wastes | Cu, Cr, V, Zn, Pb, Co, itd. | [69] |
| 7 | biohydrometallurgy biorecovery (bioprecipitation, biosorption, bioreduction, bioaccumulation) | sludges, dusts, residues, slags, red mud, and tailing wastes | | [70] |
| 8 | review methods | coal and coal combustion byproducts | REE 90–99.8% | [71] |

*4.4. Environmental Protection of Historical Waste Landfills*

Remediation of residues obtained during mining and metallurgical activities is also important from an environmental aspect. There are several studies focused on the pollution of the environment as a consequence of long-term mining and smelting activities in the West Balkan. The study [72] concluded that intensive chromium recovery and discharge of inert waste after recovery to the environment without any regularity in Albania causes numerous irreversible degradations in the environment. Another study, performed in Bosnia and Herzegovina, gives an insight into the serious soil contamination of Stavnja valley as a consequence of long-term activities in mining and processing facilities [73]. A study performed in Kosovo *, based on moss analysis, showed that mosses collected in the vicinity of processing plants have high Pb and Zn concentrations, occasionally also As and Cd, reflecting heavily on air pollution [74]. The study by Mihailovski et al. [75] investigated the content of chromium in groundwater and surface water around the industrial dump Jugohrom-Jegunove in North Macedonia and concluded that the concentration of $Cr^{6+}$ in most of the samples analyzed exceed the maximum concentration permitted according to the drinking water standards. In the same way, the study by Krivokapić et al. [76] focused on the quality of water and sediments of Skadar Lake in Montenegro in an area with a long history of overexploitation of natural resources. Similarly, the study Ranđelović et al. [7] exemplifies the importance of waste management and shows the increased concentrations of chemical elements in watersheds after the collapse of a tailing dam at an antimony mine in Serbia.

There are several mining-related hotspots in the West Balkan. These areas are heavily polluted not only by long-term mining activities, but also by lack of knowledge and/or legalization, while the disintegration of Yugoslavia has marginalized environmental man-

agement and control. According to the report by Peck et al. [21], there are three sites in Albania that pose a threat to the environment, mainly because of acid mine drainage. These tailings ponds are in Fushe Arrez, Reps, and Rreshen. In Northern Macedonia, there are 13 of the most important mining-related hotspots, 9 of which are related to mining activities and 4 are a consequence of metallurgical activities [77]. MESP, through the Environmental Protection Agency of Kosovo *, concluded that 26 out of 110 selected sensitive points were identified as potential hotspots. They are located in the municipalities of Kosovska Mitrovica, Obilić, Glogovac, Kišnica, and Novo Brdo, among others. The total area of heavily polluted regions is about 10 km$^2$, which is 0.09% of the territory of Kosovo *. Moreover, Kosovo * has neither a strategic plan nor a specific program for the treatment of hotspots [20], which further complicates the situation. According to the UNEP report [78], much has also been done in Serbia and Montenegro since 1999 to address the most serious environmental problems. Indeed, in order to solve the problems related to the remediation of hotspots in Serbia and Montenegro, as in other countries, large financial resources and extensive studies are needed. In addition, favorable policies should be developed to encourage the private sector and build a partnership for environmental management [78].

It is apparent that mining, along with industry in urban areas, fossil-fuel-fired thermal power plants, and improper disposal of waste and wastewater, is not only a problem in the West Balkan region, but it is one of the world's largest environmental issues. The proposed study wants to strive with zero-waste philosophy, which means that after extracting valuable raw materials such as CRM, precious, and base metals, and removing toxic metals, the residues can be recycled for the construction sector [40,41]. This holistic, eco-innovative approach to the recovery of valuable metals and the useful use of residues after metal recovery forms the principles of the circular economy and in this way protects the environment. Additionally, waste landfill treatment could affect the region's employment and social protection. Currently, about 4.5 million people from the West Balkan live outside their country of origin [79], while unemployment in the West Balkan countries is the highest in Europe. Locals suffer from comparatively low employment rates and a lack of opportunities for young workers, who are consequently forced to look for work outside their country, while those who stay in their country of origin often live in poverty [80]. The gross domestic product per capita (GDP) in 2020 in the region ranges between 5250 USD for Albania up to 14,100 USD for Croatia, when adjusted by purchasing power parity (PPP) [31]. Thus, the study, inter alia, also offers opportunities to improve the socioeconomic aspect of the West Balkan countries.

## 5. Conclusions

Mining waste management is currently based on linear economic thinking. However, recent political instabilities and unforeseen situations such as the COVID-19 pandemic show that mining waste as a commodity can both provide solutions to limited metal supplies and transform linear economic thinking into one that is circular. For this reason, the article addressed the list of potential secondary raw material sources with economic value in the countries of the West Balkan. It was concluded that there are 73 tons of gold, 1500 tons of silver, 3100 tons of bismuth, 310,000 tons of Cu, 260 tons of indium, 55,000 tons of molybdenum, 620,000 tons of lead, 5.3 tons of rhenium, 15,000 tons of antimony, 3700 tons of tungsten, and 970,000 tons of zinc in a total of 42 tailings piles, either from mining or from processed and metallurgical slags, which corresponds to a profit of 18,100 million USD. The study has shown that the tailings typical of the West Balkan have a significant content and quantity of valuable elements, with Serbia and Kosovo * being the areas with the greatest potential for these activities. The development of appropriate novel technologies in line with zero-waste philosophy could lead to the strengthening of the economy and increase the competitiveness of raw material supply with other European countries, while the rehabilitation of tailings would reduce environmental pollution.

**Author Contributions:** Conceptualization, R.Š. and I.R.; methodology, R.Š.; software, R.Š.; validation, R.Š. and I.R.; formal analysis, R.Š.; investigation, R.Š. and I.R.; resources, R.Š. and I.R.; data curation,

R.Š.; writ-ing—original draft preparation, B.Č.; writing—review and editing, B.Č., R.Š. and I.R.; visualiza-tion, R.Š.; supervision, R.Š. and I.R. All authors have read and agreed to the published version of the manuscript.

**Funding:** This research was partly funded by EIT RM projects: RESEERVE grant number 17029, RIS-RECOVER grant number 17128, RIS-CuRE grant number 18248 and Slovenian Research Agency research core funding number P1-0020.

**Data Availability Statement:** Data is contained within manuscript.

**Conflicts of Interest:** The authors declare no conflict of interest.

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
