# Peer review of "Mining and Metallurgical Waste as Potential Secondary Sources of Metals—A Case Study for the West Balkan Region"

_minerals, doi:10.3390/min12050547_

Round 1

Reviewer 1 Report

This is an interesting manuscript. However, there are some shortcomings/drawbacks and improvements and clarifications are needed:

  1. The aim of the manuscript: “The aim of this paper is to highlight the prospects for extracting metals from mine waste in the West Balkan” is  not scientifically sound and it needs to be clarified as well as better elaborated.
  2. It is not clear enough what is the novelty or originality of the manuscript, especially comparing with references no. 13 and 26. How does this manuscript differ from the cited publications 13 and 26?
  3. The references for the introduction chapter are awkward and need to be verified very carefully:
  1. Why the authors discussing critical raw materials in Europe do not cite the most official and reliable source of information, which is published here COM(2020) 474 - Critical Raw Materials Resilience: Charting a Path towards greater Security and Sustainability?
  2. Why do the authors discussing Sustainable Development Goals cite World Economic Forum Annual Report but not the first original source of information https://www.un.org/sustainabledevelopment/sustainable-development-goals/?
  3. Discussing the European Commission authors cite UNEP not European Commission
  1. Authors often refer to the projects: RESEERVE, RIS-CuRE, RIS-RECOVER but it is not sufficiently clear it the authors are/were participants of these projects.
  2. It is not explained (line 199 and further) how were the exact places of sampling selected? Were laboratory samples representative? How was that achieved? How were lab samples prepared for chemical analyses?
  3. “Table 5 shows the potential profit from the extraction of metals from mine waste”, if so were  CAPEX and OPEX calculated? What are their values?
  4. The word “extraction” is incorrect in the manuscript, extraction is only one step in the whole technological process of materials recovery, the better word for the whole process is recovery instead of extraction. Recovery includes: excavation, beneficiation, leaching and extraction, it should be clarified somewhere in the manuscript
  5. Enriched is a wrong word in the context of the manuscript, enriched means especially prepared/processed for example enriched uranium, it is suggested to use here rich in …. certain metals.
  6. Line 119 and further:, it is recommended to used billion USD also in second part of sentence
  7. Table 6 “Review methods” means nothing , please provide some information. The same for “industrial wastes”, what type of industry? “Industrial sludge” - what type of industry? what type oF sludge?
  8. Conclusions: “after extracting valuable raw materials such as CRM, precious and base metals and removing toxic metals, the residues can be recycled for the construction sector” Is it proven in the manuscript? Are the characteristic of the residues given in the manuscript? Conslusions shoub be fully supported by the results.
  9. Profits are not supported by CAPEX or OPEX.

Author Response

Answers to the Reviewer 1
No.    Authors answer
1.    We made a correction (L109): “The purpose of this paper is to present the chemical composition of mining waste material in West Balkan as possible sources of strategically important metals”. 
2.    We added an explanation (L113): “The contribution uses and disseminates the data presented in the not yet published article by Šajn et al [28].and for the first time shows the total profitability of metal recovery in West Balkan.”
3.    
a)    Thank you for the warning. We added this reference (L32).
b)    We made a correction (L56)
c)    We made a correction (L78)
4.    We made a correction (L200).
5.    We added a new paragraph (L229-236) and explain sample preparation in more detail. 
6.    In the study, we didn’t focus on the real cost to realize the metal recovery from the waste material. This is an overview article, which shows possibilities for further metal recovery. We added a sentence in L117.
7.    We made corrections. 
8.    We made corrections. 
9.    We made corrections. 
10.    We made corrections and added a paragraph (L576-591) with an explanation. 
11.    We made a correction (L641). 
12.    As I have already mentioned, this is an overview article, where only possibilities of metal recovery are presented, not the realization of the plan. For this step, we need to focus on every tailing dam separately, which is not the point of this article. This will be the subject of the next publication

Reviewer 2 Report

The study presented an interesting overview about the technological potential of minero-metallurgical wastes in the Balkans. With some specific corrections could be a useful reference in the future.

The number of references is adequate and it seems to cover the state of the art accordingly.

However, some citations seem to be missing in Table 1 and 3. The authors could please explain the source of these data.

In Table 2, some values are in red. The authors mentioned below that “Important values are bolded”. Did they refer to that? Why are these important values? Please, explain.

The methodology for surface area calculations must be clearly presented. Moreover, the authors must present the equipment (manufacturer and model) for the ICP analysis.

Author Response

Answers to the Reviewer 2

  1. We made a correction by adding additional columns in Tables 1 and 3. An explanation of sources and selected data is described in lines 216-227.
  2. We made a correction by changing the expression “bolded” to “coloured in red”. These values are important as they exceed the predefined values (explained in L302-306).
  3. We made corrections and explained the process of surface area estimations (L 225-227). We also add a link to the ACME laboratory, where all information can be found (L242). 

Reviewer 3 Report

Dear Authors

I want to congratulate you on a wide research program with a perspective on recommendations for real-world applications. As a practicing engineer and university staff in one person, I value academic studies for their applicability. The Present subject is vital and important because many countries face the problem of mining and industrial (metallurgical, smelting) waste and its possible processing and reuse. I appreciate any new developments in that field of knowledge.

My general and detailed comments are given below with a set of suggested references. None of the proposed papers is co-authored by me so I have no personal gain from my reference suggestions. Please do not consider them as a mandatory list but rather just like some inspiring proposals.

  1. Your contribution is quite well composed and strictly follows the IMRaD structure of scientific contribution. Abstract covers briefly the content of the study.
  2. Introductory part is very well written. It is comprehensive with regard to problem description in a global scale, but it is finally focused mainly on the Region under study. It is absolutely clear to me, but I have an impression that “State of the Art report” could cover wider region then West Balkan countries. Please develop your research to provide more international background and experiences gathered worldwide for further analyses in the last sections of your study. Usually, it wouldn't bother me, if the problem under study had just a local, Balkan importance. But your study is focused on issues of global importance (or, better to say, local problem appearing worldwide). That is why I'd suggest to widen your reference list, considering articles from other countries, based on other (maybe different) approach to the problem. That could be beneficial for the quality of your study, but also for its future citing potential, helping to focus other researchers’ attention on your work.
  3. Concerning the introductory part again. I'd recommend avoiding "group references" like [15-19], as every cited reference deserves to be introduced to the Reader, to show its impact and importance for the current study.
  4. Concerning Figures 4, 5 and 7. Please try to provide a little bigger font in axis. That will increase the readability for mature researchers.
  5. Concerning the reference list - I just made a list of papers that attracted my attention and seem to be very relevant. They refer to Czech Republic, Russia, Indonesia and Moldova. References 2 and 5 seem to be focused on possible mineral extraction (like your study), others refer also to possible use of mineral waste for production of backfill materials. Most of them are recent and open-access. Please feel free to make your own selection and/or your quick search in databases.
  1. Niemiec, D.; Duraj, M.; Cheng, X.; Marschalko, M.; Kubac, J. Selected black-coal mine waste dumps in the Ostrava Karvina region: An analysis of their potential use. IOP Conf. Ser.: Earth Environ. Sci. 2017, https://doi.org/10.1088/1755-1315/95/4/042061
  2. Rybak, J.; Adigamov, A.; Kongar-Syuryun, C.; Khayrutdinov, M.; Tyulyaeva, Y. Renewable-Resource Technologies in Mining and Metallurgical Enterprises Providing Environmental Safety. Minerals 2021, 11, 1145. https://doi.org/10.3390/min11101145
  3. Rybak, J.; Gorbatyuk, S.M.; Kongar-Syuryun, C.; Khayrutdinov, A.M.; Tyulyaeva, Y.; Makarov, P.S. Utilization of Mineral Waste: A Method for Expanding the Mineral Resource Base of a Mining and Smelting Company. Metallurgist 2021, 64, 851–861. https://doi.org/10.1007/s11015-021-01065-5
  4. Hefni, M.; Ahmed, H.A.M.; Omar, E.S.; Ali, M.A. The potential re-use of Saudi mine tailings in mine backfill: A path towards sustainable mining in Saudi Arabia. Sustainability 2021, 13, 6204. https://doi.org/10.3390/su13116204
  5. Zglinicki, K.; Szamałek, K.; Wołkowicz, S. Critical Minerals from Post-Processing Tailing. A Case Study from Bangka Island, Indonesia. Minerals 2021, 11, 352. https://doi.org/10.3390/min11040352
  6. Zglinicki, K.; Małek, R.; Szamałek, K.; Wołkowicz, S. Mining Waste as a Potential Additional Source of HREE and U for the European Green Deal: A Case Study of Bangka Island (Indonesia). Minerals 2022, 12, 44. https://doi.org/10.3390/min12010044
  7. Ponomarenko, T.; Nevskaya, M.; Jonek-Kowalska, I. Mineral Resource Depletion Assessment: Alternatives. Problems. Results. Sustainability 2021, 13, 862. https://doi.org/10.3390/su13020862
  8. Resniova, E.; Ponomarenko, T. Sustainable Development of the Energy Sector in a Country Deficient in Mineral Resources: The Case of the Republic of Moldova. Sustainability 2021, 13, 3261. https://doi.org/10.3390/su13063261

Best regards

Author Response

Answers to the Reviewer 3

  1. Dear reviewer, thank you for taking your time and making a revision of the manuscript, as well as for the lovely comment.
  2. We made a correction by adding the paragraph about background about the application of waste material (lines between 58-75).
  3. We made corrections.
  4. We made corrections.
  5. Thank you.very much for your effort. We used some of the mentioned references. The text between lines 58-75. 

Round 2

Reviewer 1 Report

The comments and suggestions for authors have been duly answered and sufficient explanations provided.

Author Response

We thank the reviewers for their time and constructive comments. 

Reviewer 2 Report

The authors improved their already relevant manuscript which is now near the  adequate condition to proceed its publication path. There is only major  unanswered issue in my point-of-view. 

I believe that the authors must consider the presentation of the equipment (manufacturer and model) used for the ICP analysis. If the website of ACME laboratory is down, or if their infrastructure is updated in the future, the readers will have access of the information in which the study was carried out. 

Author Response

Thank you very much for your comment

The http address is added: https://www.bvna.com/mining-laboratory-services

Indeed, it is important to know the equipment information but since we did the analyses in the routine laboratory, we have no that information. I was looking at their web site, but I didn’t fount that. I have to say, that I have very good experience with this laboratory, in last 30 years, I personally send them for analyses over 10.000 samples. The set of analysed samples had included the replicants and standard materials for QA/QC. In 30 years, I never complained about any given results.

Reviewer 3 Report

Dear Authors

I appreciate the work that you put to improve your paper.

I do not notice any important issues that should be subject of further corrections. I just wonder if you could provide some more recent data concerning lines 640-642 and reference [79]. Information published in 2014 were gathered probably in 2012-2013 (meaning almost 10 years ago). I understand that this information was given just to underline the necessity of development of Albania and Macedonia, but still some newer data would be appreciated.

Best regards

Author Response

Thank you very much for your comments and time for reviewing our manuscript. The suggested remarks is accepted and the paragraph is adjusted accordingly.